# Gestational and congenital syphilis across the international border in Brazil

**Leonor H. Lannoy[1], Patrícia C. Santos[2], Ronaldo Coelho[2], Adriano S. Dias-Santos[2], Ricardo Valentim[3], Gerson M. Pereira[2], Angelica E. Miranda[1] ***

**1** Post-Graduation Program in Infectious Diseases, Federal University of Espírito Santo, Vitória, Espírito Santo, Brazil, **2** Department of Chronic Diseases and Sexually Transmitted Infections, Secretary of Health Surveillance, Brazilian Ministry of Health, Brasilia, Distrito Federal, Brazil, **3** Laboratory of Technological Innovation in Health (LAIS), Federal University of Rio Grande do Norte, Natal, Rio Grande do Norte, Brazil

* amiranda.ufes@gmail.com

**Data Availability Statement:** All relevant data are within the paper and its Supporting information files. A global view of all Brazilian data are available at http://indicadoressifilis.aids.gov.br/.

## Abstract

### Background

Brazil lacks data from syphilis in its border areas. We aimed to describe the spatial and temporal distribution of acquired syphilis (AS), in pregnancy (SP) and congenital syphilis (CS) in Brazilian municipalities in the arches border contexts.

### Methods

An ecological, cross-sectional study was conducted from 2010 to 2020. The study was based on the cases of syphilis available in the Notifiable Diseases Information System (SINAN), and on the Primary Health Care Information System. The detection rates of AS and SP, and the incidence of CS were estimated, and the time series was analyzed. Data between the border arches were compared.

### Results

In 2020, data showed 7,603 cases of AS (detection rate 64.8/100,000 inhabitants), 3,960 cases of SP (detection rate of 21.6/1,000 live births) and 836 cases of CS (incidence of 4.6/ 1,000 live births) in the border region. Between 2010 and 2020, the mean annual increase of detection rate of SP was 53.4% in Brazil, 48.0% in the border region, 59.6% in the North Arch, 28.8% in the Central and 67.2% in the South. Annual variation on the incidence of CS for the same period was 31.0% in Brazil 38.4% at the border, in the North and South Arcs 18.3% and 65.7% respectively. The Central Arch showed an increase only between 2010 and 2018 (62.7%). A total of 427 (72.6%) municipalities has primary health care coverage ≥ 95% of the population. In 2019, 538 (91.8%) municipalities reported using rapid tests for syphilis, which decreased to 492 (84%) in 2020. In 2019, 441 (75.3%) municipalities reported administering penicillin, and 422 (72%) in 2020.

### Conclusion

Our data show syphilis reman problem at the Brazilian border, rates in pregnant are high. It was observed a reduction in the detection rates, SP and the incidence of CS between 2018

**Funding:** The authors received no specific funding for this work.

**Competing interests:** The authors have declared that no competing interests exist.

and 2020. Syphilis should be included on the agenda of all management levels, aiming at expanding access and quality care.

## Introduction

The Brazilian border region consists of 9 countries and the French Guiana, with 15,179 km of extension, 150 km of width and an area of 1.4 million $km^2$ (equivalent to 16.6% of the Brazilian territory). The region comprises 11 states, 588 land municipalities, two aquifers and more than 10 million inhabitants (11,733,448) [1]. Its extension has spaces with different geopolitical characteristics that involve issues related to mobility, language barriers and situations of illegality, which lead to inaccessibility and affects physical, social and mental vulnerability [2].

Brazilian borders present different levels of integration and economic development depending on international political and economic dynamics that can hamper or facilitate local socioeconomic development [3]. Cultural diversity, the political structures of municipalities, the level of economic development, among others, influence the functioning and demand of local health systems [4]. Thus, global processes increasingly determine the health of a population. In general, these territories tend to show indicators of lower income, lower well-being, conditions of backwardness and socioeconomic and environmental vulnerability [5].

Syphilis is one of the most common sexually transmitted infections (STIs), with about 7.1 million new cases worldwide in 2020 [6]. Estimates show there were more than half a million (approximately 661,000) cases of congenital syphilis (CS) worldwide in 2016, which resulted in more than 200,000 stillborn and neonatal deaths [7]. Between 2010 and 2020, in Brazil, 853,256 cases of acquired syphilis (AS), 449,981 cases of syphilis in pregnancy (SP) and 197,700 cases of CS were reported in the Notifiable Diseases Information System (SINAN). In 2020, the AS detection rate was 54.5 cases/100,000 inhabitants (115,371 cases), the SP detection rate was 21.6 cases per 1,000 live births (LB) (61,441 cases) and the CS incidence was 7.7 cases per 1,000 live births (22,065 cases) [8]. These data emphasize the infection as a national public health problem.

Syphilis remains a global challenge in several countries and in Brazil it persists as a public health problem. In 2016, syphilis was considered a serious public health problem in Brazil due to constant increases in the detection rates of AS, SP and the incidence of CS. For this reason, the Ministry of Health prepared and launched the "Agenda of Strategic Actions for the Reduction of Congenital Syphilis 2017–2019" and the following year the "Rapid Response to Syphilis Project" to implement access to the diagnosis and treatment of syphilis throughout the country [9].

Primary health care diagnoses and treats AS and SP in Brazil, with 76.1% of population coverage in 2020 [10]. A strategy to diagnose the cases is first testing for syphilis with the point-of-care test (treponemal) and then confirming with non-treponemal tests, such as the Venereal Disease Research Laboratory Test (VDRL). Penicillin G Benzathine is recommended as the first choice for treatment [11].

The AS detection rates differ between the regions of Brazil and reflect in the prevalence of SP [12]. The differences between the regions may also suggest different levels of quality of services and accessibility, which affect the diagnosis and treatment. The incidence of congenital syphilis reflects this scenario [13].

Despite the importance of the border region in Brazil and the CS as a public health problem, few studies explored the epidemiology of this infection in the region. However, the knowledge on the dissemination of syphilis in the Brazilian border and the availability of

primary health care services can help to understand the dynamics of the infection and allow control measures that meet the peculiarities of these environments. Based on it, this study was performed to describe the spatial and temporal distribution of syphilis in Brazilian municipalities in the arches border contexts.

## Methods

This is an exploratory ecological study with multiple group designs. It is a descriptive study conducted with secondary data available in the information systems of the Brazilian Ministry of Health regarding land border municipalities in Brazil.

Information from the 586 municipalities of the land border were included and the two aquifer municipalities, Lagoa dos Patos and Lagoa Mirim, in the state of Rio Grande do Sul, were excluded since they did not have a population to be analyzed. Sociodemographic variables of health conditions, procedures/treatments performed (point-of-care tests and administration of penicillin G Benzathine), dates of diagnoses, outcome of pregnancy regarding congenital syphilis and coverage of primary health care were studied.

The definition used by the Border Development Program (PDFF) [14] of the Brazilian Government was considered for analysis, which establishes that the region is divided into three arches (North, Central and South). The North Arch includes the borders of the Federative Units, Amapá, Pará, Roraima, Amazonas and Acre with 69 municipalities; the Central Arch includes Rondônia, Mato Grosso and Mato Grosso do Sul with 101 municipalities and the South Arch includes the border of Paraná, Santa Catarina and Rio Grande do Sul, with 418 municipalities (two aquifers). The aspects of productive basis and cultural identity are considered for this division to emphasize the local potentials with the relationship and articulation with neighboring countries and implement actions according to the particularities of each region.

The detection rates of AS, SP and the incidence of CS were obtained from the data available in the panel of indicators of syphilis (http://indicadoressifilis.aids.gov.br/) [15] between 2010 and 2020. The coverage of diagnostic tests for syphilis and the analysis of administration of penicillin G Benzathine by primary health care were analyzed based on the information in the Health Information System for Primary Health Care [16].

The epidemiological data of the studied infection in the three population was obtained by nationally analyzing the notification data in the SINAN and the Primary Health Care Information System (E-SUS). A descriptive analysis with frequency distribution for qualitative variables and an estimate of mean for quantitative variables were performed, as well as appropriate graphs and maps.

The mean annual increase (annual variation) of detection rates of SP and incidence of CS between 2010 and 2020 and period from 2010 to 2018, were estimated for the evaluation of detection rates. The period from 2010 to 2018, was chosen because in October 2017 the congenital syphilis definition was changed in Brazil, it was adopted the case definition proposed by the WHO [17]. The detection rates underwent logarithmic transformation and then the autoregressive Prais-Winsten model was applied, according to Antunes and Cardoso's methodology [18].

The coefficients of variation ($\beta_1$) obtained after the model was applied, which correspond to each of the periods, were used in the following formula to obtain the annual variation rates:

$$\mathbf{Rate} = [-1 + e^{\beta_1}] \times 100$$

From the coefficient of variation ($\beta_1$) and the standard error, the values of the confidence interval (CI) of the variation rates were obtained.

The significance level considered was 5%. The quantitative data were descriptively analyzed using the Statistical Package for the Social Sciences (SPSS) for Windows, version 20.0, Chicago, SPSS Inc. The time series analysis was performed using Stata for Windows, version 13.0. The program TabWin version 3.6, free software, was also used for the construction of thematic maps (http://siab.datasus.gov.br/DATASUS/).

The databases were analyzed without the identification of the subjects. It was an anonymous secondary database analysis performed after the authorization of the Brazilian Ministry of Health. The study was submitted to the Research Ethics Committee of the Center for Health Sciences of the Federal University of Espírito Santo, as recommended by resolution no. 466/2012 of the National Health Council and approved under the number 4,719,557/2021. Privacy and confidentiality were ensured at all stages of the project by codification.

## Results

In 2020, the border region reported 7,603 cases of AS (detection rate 64.8/100,000 inhabitants), 3,960 cases of SP (detection rate of 21.6/1000 LB) and 836 cases of CS (incidence of 4.6/1,000 LB). Fig 1 shows that the detection and incidence rates of these infection cases differ in the three arches.

The analysis of the sociodemographic characteristics of pregnant women with syphilis in 2020 shows a mean age of 24 years old, a mode of 20 years old, and median of 23 years old. This age categorization shows that 938 (23.7%) were between 15 and 19 years old, 2,194 (55.4%) between 20 and 29 years old and 701 (17.7%) were ≥ 30 years old. Information on schooling was reported as not informed by 871 (22%), 1,325 (34.2%) were illiterate or attended

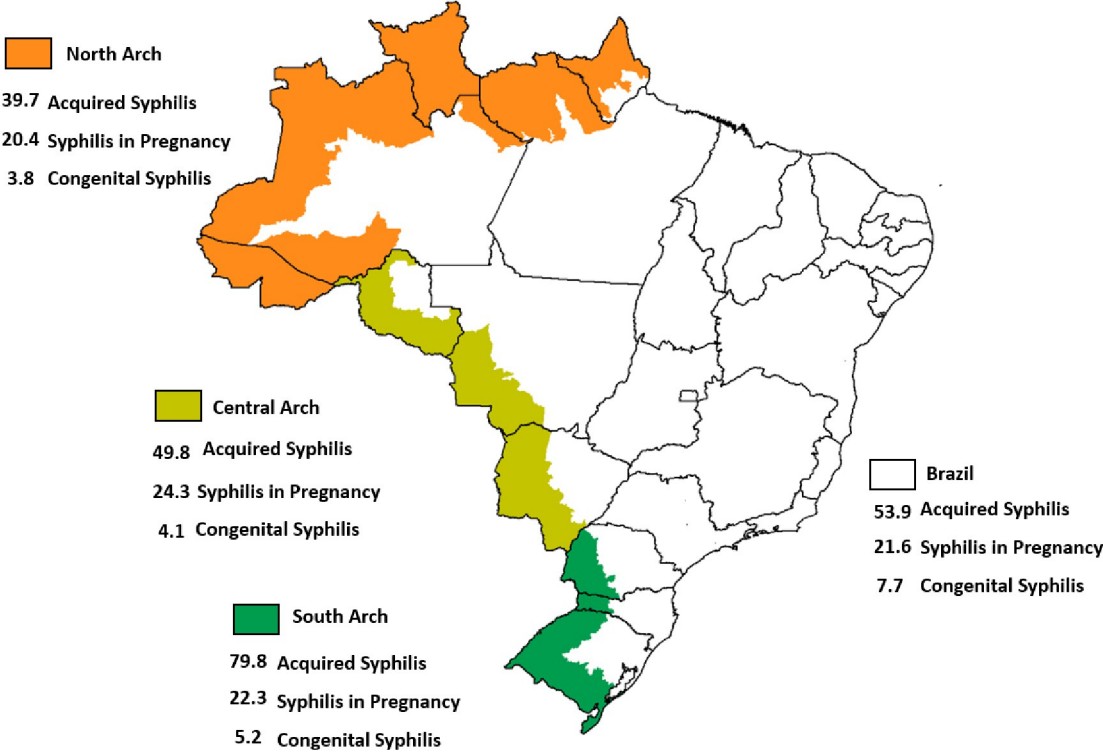

**Fig 1. Detection rate of acquired syphilis per 100,000 inhabitants, syphilis in pregnancy and incidence of congenital syphilis per 1,000 live births, according to the border arches and in Brazil, 2020.** Source: http://siab.datasus.gov.br/DATASUS/.

middle school, 1,567 (39.5%) had incomplete or complete high school and 169 (4.3%) had entered or completed higher education. Regarding race/color, 38.4% (1,520) declared themselves as white and 58.0% (2,295) as others (black, mixed race, Indigenous, Asian). We observed that the diagnosis of syphilis occurred in the first trimester of pregnancy in 1,760 (44.4%) of cases, with a greater proportion in the South Arch (52.2%), followed by the Central (40.9%) and North (32.9%) Arches. The clinical classification of syphilis was reported as not informed by 822 (20.8%) and 1,286 (32.5%) reported primary syphilis. Regarding the treatment of syphilis, 3,215 (81.1%) received adequate prescriptions based on the penicillin dose and clinical classification. In the Central Arch, however, only 762 (76.9%) pregnant women received adequate treatment. In 2020, 836 (26.8%) cases of syphilis in pregnancy resulted in congenital syphilis in the Brazilian border, with greater proportion, 30.7% (459), in the South Arch (Table 1).

**Table 1. Sociodemographic characteristics, gestational age when diagnosed with syphilis, clinical classification, treatment of reported cases of pregnant women with syphilis and proportion of outcomes, according to the border arch of residence and year of delivery 2020.**

|  | North Arch | | Central Arch | | South Arch | | Total | |
|---|---|---|---|---|---|---|---|---|
| Age group | N | % | N | % | N | % | N | % |
| Under 15 years old | 21 | 2.1 | 11 | 1.1 | 26 | 1.3 | 58 | 1.5 |
| 15 to 19 years old | 278 | 27.4 | 246 | 24.8 | 414 | 21.2 | 938 | 23.7 |
| 20 to 29 years old | 531 | 52.4 | 565 | 57.0 | 1,098 | 56.2 | 2,194 | 55.4 |
| ≥ 30 years old | 184 | 18.1 | 169 | 17.1 | 417 | 21.3 | 770 | 17.7 |
| Schooling |  |  |  |  |  |  |  |  |
| Up to Middle school | 385 | 38.0 | 358 | 36.1 | 609 | 31.2 | 1,353 | 34.2 |
| High School | 431 | 42.5 | 326 | 32.9 | 810 | 41.5 | 1,567 | 39.5 |
| Higher education | 40 | 3.9 | 37 | 3.7 | 92 | 4.7 | 169 | 4.3 |
| Not informed | 158 | 15.6 | 270 | 27.2 | 443 | 22.7 | 871 | 22.0 |
| Race/color |  |  |  |  |  |  |  |  |
| White | 57 | 5.6 | 254 | 25.6 | 1,209 | 61.8 | 1,520 | 38.4 |
| Other | 924 | 91.1 | 712 | 71.8 | 659 | 33.7 | 2,295 | 57.9 |
| Not informed | 33 | 3.3 | 25 | 2.5 | 87 | 4.5 | 145 | 3.7 |
| Gestational age of diagnosis |  |  |  |  |  |  |  |  |
| First trimester | 334 | 32.9 | 405 | 40.9 | 1,021 | 52.2 | 1,760 | 44.4 |
| Second trimester | 242 | 23.9 | 210 | 21.2 | 395 | 20.2 | 847 | 21.4 |
| Third trimester | 315 | 31.1 | 346 | 34.9 | 456 | 23.3 | 1,117 | 28.2 |
| Gestational age not informed | 123 | 12.1 | 30 | 3.0 | 83 | 4.2 | 236 | 6.0 |
| Clinical classification |  |  |  |  |  |  |  |  |
| Primary syphilis | 344 | 33.9 | 264 | 26.6 | 678 | 34.7 | 1,286 | 32.5 |
| Secondary syphilis | 52 | 5.1 | 47 | 4.7 | 80 | 4.1 | 179 | 4.5 |
| Tertiary syphilis | 96 | 9.5 | 156 | 15.7 | 107 | 5.5 | 359 | 9.1 |
| Latent syphilis | 331 | 32.6 | 384 | 38.7 | 599 | 30.6 | 1,314 | 33.2 |
| Not informed | 191 | 18.8 | 140 | 14.1 | 491 | 25.1 | 822 | 20.7 |
| Treatment |  |  |  |  |  |  |  |  |
| Adequate | 836 | 82.4 | 762 | 76.9 | 1,617 | 82.7 | 3,215 | 81.1 |
| Inadequate | 138 | 13.6 | 97 | 9.8 | 163 | 8.3 | 398 | 10.1 |
| Not performed/ Not informed | 40 | 3.9 | 132 | 13.3 | 175 | 9.0 | 347 | 8.8 |
| Outcome of children exposed to syphilis |  |  |  |  |  |  |  |  |
| CS—yes | 188 | 18.5 | 189 | 19.1 | 459 | 30.7 | 836 | 26.8 |
| CS—no | 826 | 81.5 | 802 | 80.9 | 1,496 | 69.3 | 3,124 | 73.2 |
| **Total** | **1,014** | **100%** | **991** | **100%** | **1,955** | **100%** | **3,960** | **100%** |

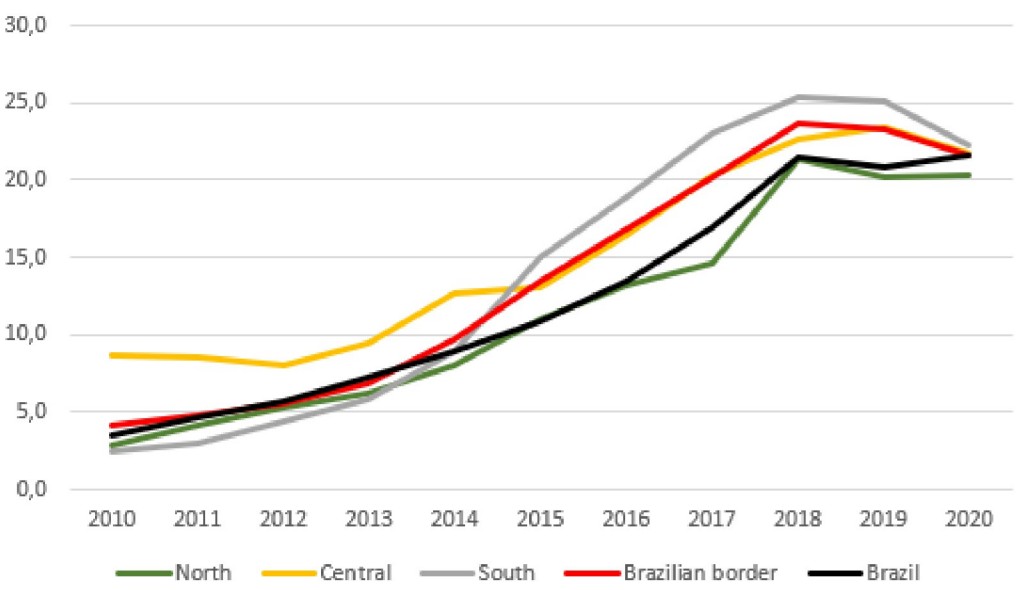

| Arches | 2010-2020 | | 2010-2018 | | Increment 2018-2020 |
|---|---|---|---|---|---|
| | rate(%) | IC95% | taxa(%) | IC95% | (%) |
| North | 59,6 * | (43.6;77.3) | 72,8 * | (64.8;81.3) | -5% |
| Central | 28,8 * | (17.6;41.0) | 34,7 * | (20.4;50.7) | -4% |
| South | 67,2 * | (32.1;111.5) | 102,0 * | (76.9;130.7) | -12% |
| Brazilian border | 48,0 * | (27.8;71.4) | 67,9 * | (55.0;81.8) | -9% |
| Brazil | 53,4 * | (39.5;68.6) | 66,3 * | (63.5;69.1) | 0% |

* p-valor < 0,05

**Fig 2. Detection rate of syphilis in pregnancy per 1,000 live births, 2010–2020 and mean annual variation of the detection rate in the three border arches and in Brazil, during 2010 to 2018, and increment rate 2018–2020.**

Fig 2 shows the mean annual increase in the detection rates of SP between 2010 and 2020. The number increased during this period in Brazil (53.4%). The mean annual increase of detection rate of syphilis in pregnancy was 59.6% in the North Arch, 28.8% in the Central and 67.2% in the South. In the analysis of the interrupted time series, the period (2010–2018) shows an increase in the three arches, North (72.8%), Central (34.7%) and South (102.0%) and in Brazil (66.3%). The increment rate was stationary between 2018 and 2020 in Brazil and decreased in the border (-9.0%).

The analysis of the annual variation of incidence of CS between 2010 and 2020 shows an increase in the incidence in Brazil (31.0%) and in the North (18.3%) and South (65.7%) Arches. The Central Arch showed an increase only between 2010 and 2018 (62.7%). The interrupted series also shows an increase in the South Arch (103.3%), Central (62,7%), North (23,4%) and in Brazil (45.8%) between 2010 and 2018. The increment of incidence of CS, between 2018 and 2020, is negative in the border region (-26%) and Brazil (-14%), being more accentuated in the central arch (-43%) (Fig 3).

The reduction in the detection rates of AS (-29.4%), SP (-7.3%) and the incidence of CS (-16.4%) in the region, between 2019 and 2020, show a better control of syphilis in the country.

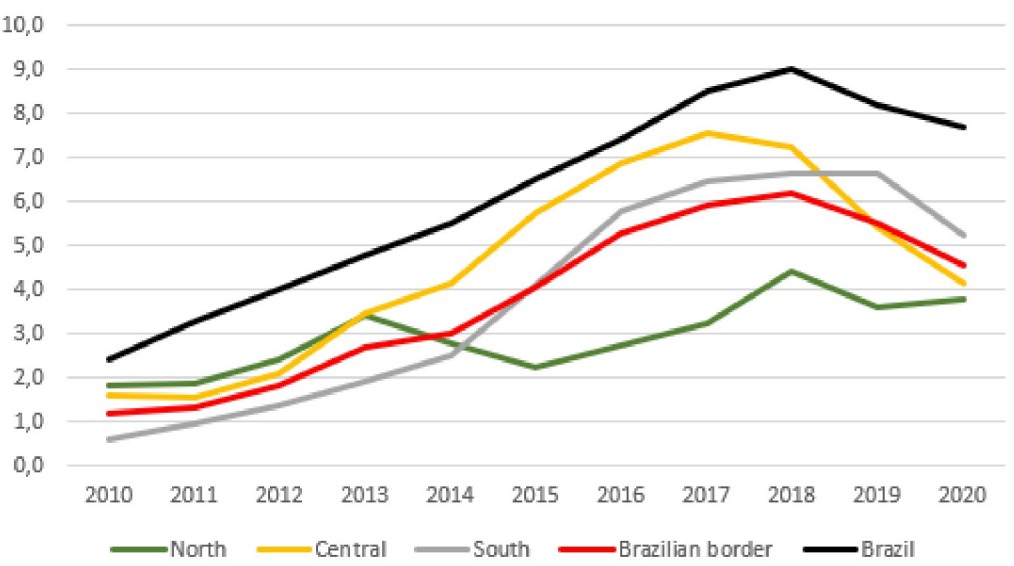

| Arches | 2010-2020 | | 2010-2018 | | Increment 2018-2020 |
|---|---|---|---|---|---|
| | taxa(%) | IC95% | taxa(%) | IC95% | (%) |
| North | 18,3 | * (7.4;30.2) | 23,4 | * (5.4;44.4) | -14% |
| Central | 26,3 | (-6.8;71.1) | 62,7 | * (38.3;91.4) | -43% |
| South | 65,7 | * (23.8;121.8) | 103,3 | * (72.0;140.3) | -21% |
| Brazilian border | 38,4 | * (10.7;73.0) | 66,7 | * (51.3;83.7) | -26% |
| Brazil | 31,0 | * (12.4;52.5) | 45,8 | * (33.9;58.7) | -14% |

* p-valor < 0,05

**Fig 3. Incidence rate of congenital syphilis per 1,000 live births, 2010–2020 and mean annual variation of incidence, in the three border arches and in Brazil, during 2010 to 2020, 2010 to 2018, and increment rate 2018–2020.**

Data from the Primary Health Care Information System (SISAB) show 538 (91.8%) municipalities reported performing point-of-care test for SP health care units in 2019. This number decreased to 492 (84%) municipalities in 2020, with the North Arch showing a greater proportion of municipalities that reported this information (96%). In 2019, 441 (75.3%) border municipalities reported administering penicillin to treat syphilis in primary health care units, and 422 (72%) reported this information in 2020. Regarding the notification of AS in 2019, 80% (469) of municipalities reported at least one case of AS, and in 2020, 71.2% (417) of municipalities reported any case in the SINAN. The North Arch had the highest proportion of this information in the SISAB (rapid test 96% and administration of penicillin 81%) and reported AS in 2020 (83%) (Fig 4).

The analysis on the primary health care coverage shows that 72.6% (427 municipalities) have a population coverage $\geq$ 95%. The Central Arch has the lowest proportion of municipalities with this coverage, 59% (58 municipalities), which is close to that of the North Arch, 61% (43 municipalities). Syphilis in pregnancy was reported in 61.4% (361) of border municipalities and 27.8% (164) have a detection rate $\geq$ national rate (21.6/1000 LB). Most municipalities, 67.5% (397), did not report cases of congenital syphilis in 2020 and 14.3% (84 municipalities) have an incidence of congenital syphilis $\geq$ 7.7/1000 LB (Fig 5).

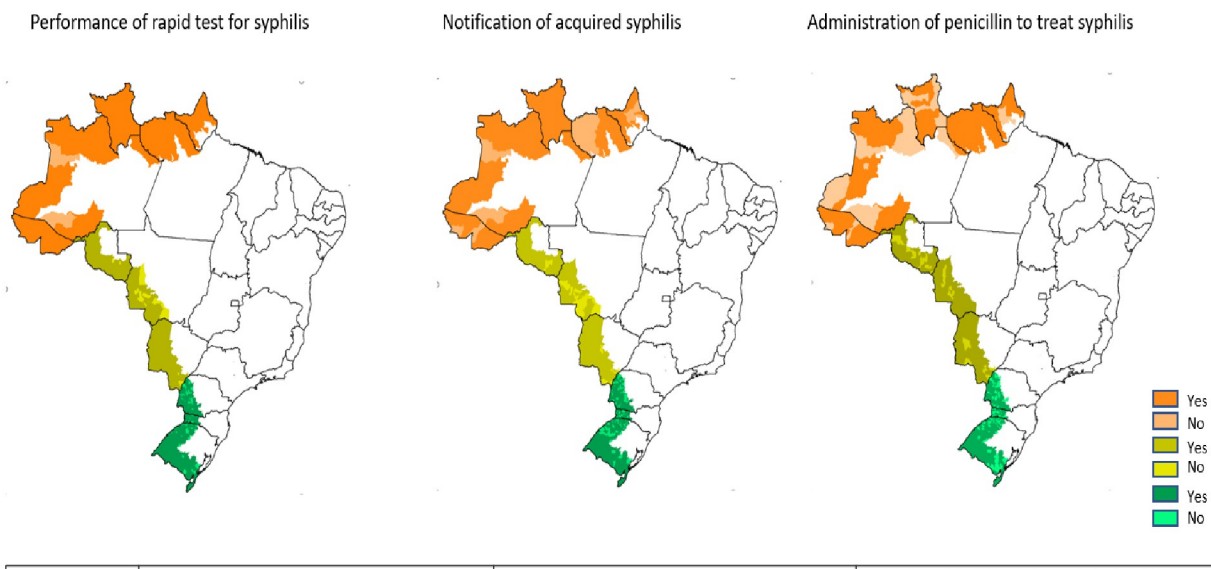

**Fig 4. Proportion of municipalities in the border that reported point-of-care test for syphilis, notified acquired syphilis and administered penicillin to treat syphilis in primary health care units, 2020.** Source: http://siab.datasus.gov.br/DATASUS/.

## Discussion

The results of our study showed that the Brazilian border region in 2020, compared to the national data, had a higher detection rate of AS, an equal detection rate of SP and a lower

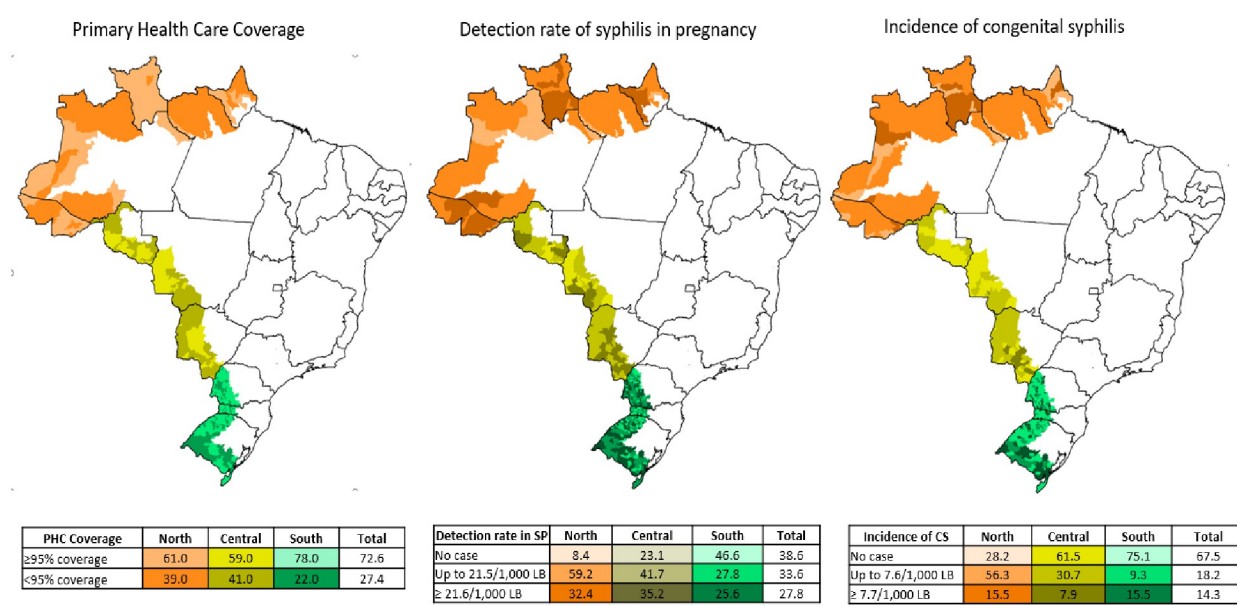

**Fig 5. Proportion of municipalities with a primary health care coverage ≥ 95%, detection rate of syphilis in pregnancy, incidence of congenital syphilis, according to the border arch, 2020.** Source: http://siab.datasus.gov.br/DATASUS/.

incidence of CS [15]. The reduction in the detection rates of AS, SP and the incidence of CS in the region, between 2019 and 2020, show a better control of syphilis in the country. Other study showed the same tendency in the country [12]. However, we cannot measure how the COVID-19 pandemic affected these data. The pandemic required a greater mobilization of health care providers, which resulted in a change in the attention that used to be directed to other diseases [19].

Despite the number of plans and initiatives to develop the border region [14, 20, 21], it faces major challenges to advance on public policies [22]. Data from the Institute for Applied Economic Research (IPEA) (2016) confirm that the border municipalities of Brazil have the worst rates of income and schooling of their respective states, and suffer from illegal immigration, currency evasion, land conflicts, smuggling and many types of trafficking (weapons, drugs, people) [23, 24]. These characteristics can affect health care and disease control.

The cross-border flow of people and the seek for services increase the vulnerability of municipalities to the entry of infectious agents, which affect the financing of health actions and services [24]. In turn, the increase of transmission of STIs, including HIV also relates to the mobility of the population associated with multiple partnerships and sexual work in these territories [25]. We must emphasize that syphilis remains endemic in Latin American, Asian and African countries with limited resources, due to the difficulty of early diagnosis and treatment, which enable its dissemination [26].

Our data showed that pregnant women with syphilis in the border region were young, non-white and with low schooling. The distribution was different depending on the analyzed arc, as the data followed the distribution of the Brazilian population, and a greater number of women who self-reported white were from the southern arc. The northern arc had the lowest percentage, where there is a greater number of indigenous people. These data are in agreement with the sociodemographic characteristics described in other studies [13, 27–29]. However, the expansion of primary health care in Brazil led to an increase in prenatal coverage and the coverage of prenatal screening exams, a measure that expanded access and service as well as improved the diagnosis and notification of syphilis cases in the country [30–32].

The notification on the staging of the clinical phases of syphilis showed that more than 30% were reported as primary syphilis, 9% as tertiary syphilis and 21% as not informed. Since most cases in pregnant women are asymptomatic, diagnosed by serological tests in the latent form of infection [33], these data may be classification errors and provide evidence of the need to continuously train health care providers to improve the possibility of adequate treatment and the quality of surveillance data in the border region [30].

The time series shows that, between 2010 and 2020, the detection rates of SP and the incidence of CS tended to increase in the border arches and in Brazil, except for the incidence of CS in the Central Arch, which strongly increased until 2017 with subsequent strong reduction between 2018 and 2020. Both SP and CS rates show a reduction as of 2018, despite the SP rate being stationary in Brazil. It is important to note that the reduction in the incidence of CS may also be related to the change in the case definition for surveillance purposes, which as of September 2017, became more specific, and excluded the condition of treatment of the pregnant woman's partner from the case definition [9].

The knowledge on the epidemiological situation of syphilis in countries bordering Brazil has been an issue in recent years due to the difference of surveillance and notification systems of the infection between the countries, as well as the priorities and available resources. This situation impairs the measuring and/or comparison between the rates of AS and SP, the vertical transmission of the infection, and the effect on local health systems related to the flows of people in the border region countries [34–40].

CS is the result of the transmission of the spirochete of the *Treponema pallidum* from the infected pregnant woman's bloodstream to the conceptus by transplacental route or, occasionally, by direct contact with the lesion at the time of delivery [33, 41]. Timely diagnosis and adequate treatment for the pregnant woman can prevent the infection, which is an essential indicator of the quality of prenatal care [13, 42]. The vertical transmission of syphilis is high and can reach almost 100% in recent forms of the infection without treatment [38]. The outcome rate of CS in the border region in 2020 was 26.7% and 35.9% in Brazil. A national hospital-based study found 34.3% (95% CI: 24.7–45.4) vertical transmission rate of syphilis [43], similar to the result of our study. This proportion of outcome is influenced by the increase in the capacity to diagnose and notify SP and the possibility to diagnose CS.

Regarding the availability of point-of-care tests for syphilis, we observed that 16% of municipalities in the border region did not report access to them in primary health care units and 28.8% did not report the administration of penicillin to treat syphilis. We must consider that border municipalities are mostly small, of low demographic density and remotely located [1]. In turn, a study with data from the National Program for Improvement of Access and Quality in Primary Health Care (PMAQ-AB) identified an improvement in the diagnoses of syphilis (treponemal and/or non-treponemal tests) in the municipalities evaluated, which enables the increase in the capacity to identify people with syphilis, and that about a third of the teams did not offer a rapid test, requiring an expansion for this measure [44].

SP is diagnosed and treated in primary health care in Brazil and since only 70% of border municipalities have primary health care coverage for 95% of the population, this care network must be strengthened to provide a better quality care [7, 29, 43]. Despite the improvements in the Brazilian Unified Health System (SUS), the control of CS, which is based on the treatment of SP, remains a challenge in Brazil and emphasizes the vulnerability in prenatal care, since it is essential to monitor access and quality of primary health care [27, 43, 44].

Our study also allowed essential analyses within each of the border arches, we emphasize below what we have found in each of them. In the North Arch, bordering the French Guiana, Suriname, Guyana, Venezuela, Colombia and Peru, despite the historical series of this territory always showing a detection rate of SP and incidence of CS, its rates were lower than in the other arches and the national mean rate. Compared to the Central and South Arches, it showed a higher proportion of pregnant women with syphilis under 19 years old (29.5%) and who studies up to middle school (38.0%), a lower proportion of detection of the infection diagnosed in the first trimester of pregnancy (32.9%) and a high proportion of primary syphilis among pregnant women (33.9%). However, it is the arch that shows the lowest proportion of congenital syphilis outcome among children exposed to syphilis (18.5%). The great territorial extension, the small population, the history of isolation from national centers and the rich cultural, ethnic and linguistic diversity [3] can influence access and adherence to health care services, as well as create care voids and hinder the identification of cases and early diagnosis.

The Central Arch borders Bolivia and Paraguay and has some of the twin cities with greater sociocultural integration and has consolidated urban centers. Local integration today is characterized by the dependence of foreign urban centers on Brazilian municipalities [1, 3, 5]. This arch shows high detection rates of SP since the beginning of the historical series and the proportion of CS outcomes is 19.1%. A total of 40.9% of the diagnoses of syphilis are made in the first trimester of pregnancy and it has the highest proportion of pregnant women with latent syphilis (38.7%). However, it is the region with the lowest proportion of pregnant women adequately treated (76.9%) and a high proportion of notifications without reporting the treatment used for syphilis and/or not performed (13.1%). The incidence of CS decreased of 43% between 2018 and 2020.

The South Arch, bordering Paraguay, Argentina and Uruguay, is the one with the highest level of development, higher urban density, with most twin cities and the highest level of regional integration based on health care due to initiatives, such as Mercosul and Unasul [3, 20]. This region shows a lower proportion of pregnant women with syphilis up to 19 years old (22.5%) and women who attended up to middle school (31.2%), a higher proportion of diagnosis in the first trimester of pregnancy (52.2%), but shows a higher proportion of pregnant women diagnosed with primary syphilis (34.7%) and no information regarding this issue (25.1%). The region has the highest proportion of CS as outcome (30.7%).

Our analysis corroborates studies that show a relationship between the prevalence of syphilis and municipalities with higher demographic densities, because they show social and health determinants that enable the dissemination of the infection and, despite offering a wider health care service, it maintains social disparities and the populations' difficulties to access the services [5, 45].

Among the limitations of our study, we mention the use of secondary data without independent validation, which comes from health administration and can be biased and problems related to under-reported cases. Another limitation is due to the design of an ecological study, which does not allow direct interpretations of individual results. The descriptive approach is limited to univariate analyses, not adjusted by many risk factors and their interactions, nor by spatial data structure, which would be possible in a more complex analytical approach. However, syphilis is of compulsory notification and information systems are the biggest source of data to perform this evaluation. And once the data were analyzed by region, the distribution of the possible underreporting dissipates less heterogeneously. Besides, these cases of syphilis are in national agreements that involve accountability and financing for states and municipalities, which expects a good data coverage.

Our data show a reduction in the detection rates of AS, SP and the incidence of CS between 2018 and 2020. This reduction needs to be evaluated in another series in the coming years, due to the pandemic and the change in the case definition, it may not be attributed only to the improvement in care. It shows that strategies focused on syphilis should be included on the agenda of all management levels, aiming at expanding access and improving quality care. We expect with our analysis to contribute to advance in the implementation of public health policy for each arch region, based on the understanding of the political, symbolic and social power relationships in which each region was formed [46]. The results showed a good performance of pregnancy care and low CS index in the border region; however, the infection must be a priority on the agenda of all management levels, aiming at expanding access and quality care. Border regions represent unique spaces for regional development and integration and are characterized by the intense sociocultural and economic exchange. Few information available on the epidemiology of STIs in these areas limit the implementation of public policies. The solutions to address public health challenges depend on bilateral relationships. The understanding on the access to health services and the behavior of diseases in the region can contribute to the well-being of the community in the region and to specific policies for the citizens on both sides.

## Supporting information

**S1 Data. Study database tables frontiers.**
(XLSX)

## Author Contributions

**Conceptualization:** Leonor H. Lannoy, Ronaldo Coelho, Ricardo Valentim, Gerson M. Pereira, Angelica E. Miranda.

**Data curation:** Adriano S. Dias-Santos.

**Formal analysis:** Leonor H. Lannoy, Patrícia C. Santos.

**Funding acquisition:** Ricardo Valentim, Angelica E. Miranda.

**Investigation:** Leonor H. Lannoy, Adriano S. Dias-Santos, Angelica E. Miranda.

**Methodology:** Leonor H. Lannoy, Patrícia C. Santos, Ronaldo Coelho, Gerson M. Pereira.

**Supervision:** Gerson M. Pereira, Angelica E. Miranda.

**Validation:** Ricardo Valentim, Angelica E. Miranda.

**Writing – original draft:** Leonor H. Lannoy, Ronaldo Coelho, Angelica E. Miranda.

**Writing – review & editing:** Leonor H. Lannoy, Patrícia C. Santos, Ronaldo Coelho, Adriano S. Dias-Santos, Ricardo Valentim, Gerson M. Pereira, Angelica E. Miranda.

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
