## [Decision Letter · Decision Letter 0]

5 Jul 2022

PONE-D-22-16035Gestational and congenital syphilis across the international border in Brazil Syphilis in Brazilian land bordersPLOS ONE

Dear Dr. Miranda,

Thank you for submitting your manuscript to PLOS ONE. After careful consideration, we feel that it has merit but does not fully meet PLOS ONE’s publication criteria as it currently stands. Therefore, we invite you to submit a revised version of the manuscript that addresses the points raised during the review process.

We look forward to receiving your revised manuscript.

Kind regards,

Everton Falcão de Oliveira, Ph.D

Academic Editor

PLOS ONE

Journal Requirements:

5. Please amend your list of authors on the manuscript to ensure that each author is linked to an affiliation. Authors’ affiliations should reflect the institution where the work was done (if authors moved subsequently, you can also list the new affiliation stating “current affiliation:….” as necessary).

6. We note that Figures 1, 4-5 in your submission contain [map/satellite] images which may be copyrighted. All PLOS content is published under the Creative Commons Attribution License (CC BY 4.0), which means that the manuscript, images, and Supporting Information files will be freely available online, and any third party is permitted to access, download, copy, distribute, and use these materials in any way, even commercially, with proper attribution. For these reasons, we cannot publish previously copyrighted maps or satellite images created using proprietary data, such as Google software (Google Maps, Street View, and Earth). For more information, see our copyright guidelines: http://journals.plos.org/plosone/s/licenses-and-copyright.

a. You may seek permission from the original copyright holder of Figures 1, 4-5 to publish the content specifically under the CC BY 4.0 license.  

a. If you are unable to obtain permission from the original copyright holder to publish these figures under the CC BY 4.0 license or if the copyright holder’s requirements are incompatible with the CC BY 4.0 license, please either i) remove the figure or ii) supply a replacement figure that complies with the CC BY 4.0 license. Please check copyright information on all replacement figures and update the figure caption with source information. If applicable, please specify in the figure caption text when a figure is similar but not identical to the original image and is therefore for illustrative purposes only.

Reviewers' comments:

Reviewer's Responses to Questions

**Comments to the Author**

1. Is the manuscript technically sound, and do the data support the conclusions?

Reviewer #1: Yes

Reviewer #2: Partly

Reviewer #3: Partly

2. Has the statistical analysis been performed appropriately and rigorously? 

Reviewer #1: Yes

Reviewer #2: Yes

Reviewer #3: Yes

3. Have the authors made all data underlying the findings in their manuscript fully available?

Reviewer #1: No

Reviewer #2: Yes

Reviewer #3: Yes

4. Is the manuscript presented in an intelligible fashion and written in standard English?

Reviewer #1: Yes

Reviewer #2: Yes

Reviewer #3: Yes

5. Review Comments to the Author

Reviewer #1: Dear Editor,

In this manuscript, the distribution of acquired syphilis, syphilis in pregnancy, and congenital syphilis was analyzed at the Brazilian border with other nine countries from 2010 to 2020. The results show that the number of cases for all three variables is increasing throughout the Brazilian border area. This is nothing new, as the number of syphilis cases has increased dramatically worldwide in recent years. However, given the lack of data on developing country borders, I believe that this manuscript should be considered for publication after some modifications as described below:

Abstract:

I suggest inserting a space between several words: pregnancy (SP), syphilis (CS), syphilis (AS), Brazil(38.4%), North(18.3%), South(65.7%), 427(72.6%), 441(75.3%), 422(72%).

Introduction:

- Why "Discussions about the border have intensified in recent decades due to the regionalization processes"?

- About "Despite the importance of the border region in Brazil and the congenital syphilis as a public health problem, few studies explored the epidemiology of this infection in the region." Are AS and SP a public health problem in Brazil? Or not?

Results:

- Note Figures 2 and 3. In the text below the figure, words are underlined. What does "2010 a 2020" mean?

- "Syphilis in pregnancy was reported in 61.4% (361) of border municipalities and 27.8% (164) have a detection rate ≥ national rate (21.6/1000 LB). Most municipalities, 67.5% (397), did not report cases of congenital syphilis in 2020 and 14.3% (84 municipalities ) have an incidence of congenital syphilis ≥ 7.7/1000 LB (Figure 5)." Define LB.

Discussion:

- Italianize Treponema pallidum: "CS is the result of the transmission of the spirochete of the Treponema pallidum from the infected pregnant woman's ..."

Reviewer #2: The manuscript is of relevance, it points out updated aspects with the syphilis theme. However, it presents internal and external biases. Title: adequate; Abstract: incomplete with the need for adaptation of items as presented throughout the text; Introduction: presents issues that belong to another topic such as the research scenario, the knowledge gap is fragile, without articulation with the uniqueness of the research. Methods: The research design is inadequate. According to Morgenstern (2011), ecological studies can be divided into: multiple group designs (exploratory and etiological), time trend designs (exploratory and etiological), mixed designs (between multiple groups and time trend, but also with sub-classifications between exploratory and etiological). By the detailing of the methods, the study is more articulated with an exploratory ecological study with multiple group designs. Results: are very well designed, but not validated with explanatory tests. The statistics are simple and make the text uncompetitive. Discussion: well elaborated with the inclusion of limitations. However, limited to the results, which in fact is not inconsistent, but does not address aspects related to the health model, the treatment of partners who are in another border country, the mobility of the standardization of treatment and availability of supplies among other aspects that interfere in the occurrence of the grievance in all its scopes, acquired syphilis, in pregnant women and congenital. I hope that the comments provided will help you with the publication elsewhere or to resubmit after adaptations.

MORGENSTERN, Hal. Ecologic Study. In: ROTHMAN, Kenneth L.; GREENLAND, Sander; LASH, Timothy L. Modern Epidemiology, 3rd Edition, 2008.

Reviewer #3: Syphilis represents a highly relevant issue globally. It remains a global challenge, the second most commonly reported STI. In Brazil, syphilis persists as a public health problem, particularly due to limited access to timely diagnosis and treatment, as well as limited monitoring of cases in the Unified Health System health care network, especially in Primary Health Care. The challenge is amplified when the critical political and institutional moment of the country is recognized. One of the great challenges has been to implement these health care actions integrated with surveillance and control, ensuring wide access to diagnosis, treatment, and monitoring in the Primary Health Care setting. These aspects are even more critical in border areas in the context of South America.

There is great variation in the operational performance of disease control in the country. This variation has been associated with operational factors such as access to testing via rapid tests, but also to the lower use of condoms, the reduced use of penicillin in routine PHC, and the period in which there was a shortage of the drug. These aspects should have been better described in the manuscript.

Moreover, both acquired syphilis, syphilis in pregnant women, and congenital syphilis are compulsorily notifiable diseases in the country but have registered systematic under-reporting, which compromises health planning actions, despite the improvement over time. This aspect should have been studied in depth.

It is important to bring the impacts related to congenital syphilis to society.

For a broader look at the real epidemiological and operational situation of syphilis control, the analyses should include the quality of prenatal care in the public and private sectors. This aspect should be discussed.

Expand the debate on the ethnic-racial clippings performed, as differentials in the three arches analyzed.

Include the debates on recently published articles:

Saes MO, Duro SMS, Gonçalves CS, Tomasi E, Facchini LA. Assessment of the appropriate management of syphilis patients in primary health care in different regions of Brazil from 2012 to 2018. Cad Saude Publica. 2022 May 16;38(5):EN231921. doi: 10.1590/0102-311XEN231921. PMID: 35584428.

Ramos AN Jr. Persistence of syphilis as a challenge for the Brazilian public health: the solution is to strengthen SUS in defense of democracy and life. Cad Saude Publica. 2022 May 16;38(5):PT069022. doi: 10.1590/0102-311XPT069022. PMID: 35584431.

Reiterate the relevance of PMAQ-AB as an innovative and useful action to induce quality improvement of PHC in SUS. This program was interrupted by the federal government in 2019 within the process of deconstruction of public health policies, having been replaced by the PREVINE Brazil Program. The authors need to discuss the effect of this change, implying a considerable setback in the process of evaluation and financing of PHC.

For the abstract, it is recommended to qualify the description of border arcs, to better situate the analyzed scenarios. The objectives of the study should have been clearly listed, as in the introduction. Thus, the objectives of the study are partially articulated with a clear testable hypothesis stated. In principle, the focus is on the spatial and temporal description in Brazilian municipalities in border contexts. The conclusions of the abstract as well as of the manuscript should be adjusted to this perspective: the epidemiological and operational patterns of syphilis control are not satisfactory. The way it is described gives the wrong message of assumed control.

The introduction needs to be enhanced with better-contextualized data in operational and epidemiological terms of syphilis control in the country and in border areas, especially over the period 2010 to 2020. In Brazil, the changes in the definition of syphilis case for compulsory notification purposes delimited the temporal scope of the study. It is important to signal which changes were undertaken and their impacts, particularly the change in the definition of appropriate treatment of pregnant women with syphilis, excluding as a criterion the concomitant treatment of the sexual partner, in terms of the sensitivity/specificity of the case definition criteria.

In the introduction, the authors should be clearer when referring to the fact that discussions about the border have intensified in recent decades due to regionalization processes, including more consistent references.

The description of the study design needs to be qualified. In principle, the study design is appropriate to address the possible objective.

In addition, a detailed map of the study area could have been presented, while in the text, the indication of the territorial and population, as well as the economic relevance of this territorial cut-out adopted in the study.

It is recommended to detail the scope of the syphilis indicator panel with its linked databases, as well as the Health Information System for Primary Health Care, clearly demonstrating the role of care and surveillance and to what extent the interfaces between these systems. The population is clearly described and appropriate for the hypothesis being tested.

Qualify correctly and better the reference in the text to SPSS version 20.0 and TabWin (version???), according to the developers' specification. The correct statistical analysis is used to support conclusions.

There are concerns about ethical requirements being met.

The results are clearly and completely presented. The figures (Tables, Images) are of sufficient quality for clarity.

In the discussion, the reduction seen in the last two years analyzed does not allow one to clearly establish that there was a reduction in the detection rates analyzed. The limitations of the analysis are clearly described.

The authors discuss partially how these data can be helpful to advance our understanding of the topic under study.

The public health relevance is addressed. However, the conclusions are partially supported by the data presented.

6. PLOS authors have the option to publish the peer review history of their article (what does this mean?). If published, this will include your full peer review and any attached files.

Reviewer #1: **Yes: **Fred Luciano Neves Santos

Reviewer #2: No

Reviewer #3: No

---

## [Author Response · Author response to Decision Letter 0]

16 Aug 2022

Response to comments to the Journal Requirements and reviewers

Journal Requirements:

RESPONSE: We reviewed the document and organized it according to the requirements

RESPONSE: Thank you for this comment, we clarified it in the text. The need for informed consent was waived by the ethics committee as this is a study used public and anonymous secondary data. Privacy and confidentiality were ensured at all stages of the project by codification performed before we had access to data. The databases were analyzed without the identification of the subjects. This study was conducted with the authorization of the Department of Chronic Diseases and Sexually Transmitted Infections of the Brazilian Ministry of Health. The study was submitted to the Research Ethics Committee of the Center for Health Sciences of the Federal University of Espírito Santo, as recommended by resolution no. 466/2012 of the National Health Council and approved under the number 4,719,557/2021. 

RESPONSE: The study did not receive any funds. We used secondary databases, and it was part of a PhD thesis. The student was responsible for the analysis and was supervised by the mentors.

RESPONSE: The funders had no role in study design, data collection and analysis, decision to publish, or preparation of the manuscript.

RESPONSE: No author received funds for this project.

RESPONSE: We included this statement in the manuscript. The authors received no specific funding for this work.

RESPONSE: We included the statement as requested. The Ministry of Health responsible for the program signed it. Amended statements are attached 

RESPONSE: All data are available at: 

The data are accessible in: 

Border municipalities: 

https://geoftp.ibge.gov.br/organizacao_do_territorio/estrutura_territorial/municipios_da_faixa_de_fronteira/2020/Municipios_da_Faixa_de_Fronteira_2020.xls

Epidemiological indicators: http://indicadoressifilis.aids.gov.br/

Primary care indicators: https://sisab.saude.gov.br/

Primary care coverage: 

https://egestorab.saude.gov.br/paginas/acessoPublico/relatorios/relHistoricoCoberturaAB.xhtml

RESPONSE: We included the database used for the project analysis.

RESPONSE: There are no ethical restrictions for database sharing. It was included in the submitted documents. 

5. Please amend your list of authors on the manuscript to ensure that each author is linked to an affiliation. Authors’ affiliations should reflect the institution where the work was done (if authors moved subsequently, you can also list the new affiliation stating “current affiliation:….” as necessary).

RESPONSE: We updated it and the statement reflects the information we provide in your cover letter.

6. We note that Figures 1, 4-5 in your submission contain [map/satellite] images which may be copyrighted. All PLOS content is published under the Creative Commons Attribution License (CC BY 4.0), which means that the manuscript, images, and Supporting Information files will be freely available online, and any third party is permitted to access, download, copy, distribute, and use these materials in any way, even commercially, with proper attribution. For these reasons, we cannot publish previously copyrighted maps or satellite images created using proprietary data, such as Google software (Google Maps, Street View, and Earth). For more information, see our copyright guidelines: http://journals.plos.org/plosone/s/licenses-and-copyright.

RESPONSE: We included the written permission from the copyright holder. The responsible director in the MoH signed the document.

As we said before, Tabwin/Tabnet is an application provided by the Ministry of Health of Brazil, with the purpose of enabling the analysis of information at the various points of the care network. This program has a public license for free and free use, available at: https://datasus.saude.gov.br/transferencia-de-arquivos/

a. You may seek permission from the original copyright holder of Figures 1, 4-5 to publish the content specifically under the CC BY 4.0 license. 

RESPONSE: We included the written permission from the copyright holder.

RESPONSE: We included the written permission from the copyright holder We contacted them, and the authorization is attached.

RESPONSE: We uploaded the content permission form with our submission. 

a. If you are unable to obtain permission from the original copyright holder to publish these figures under the CC BY 4.0 license or if the copyright holder’s requirements are incompatible with the CC BY 4.0 license, please either i) remove the figure or ii) supply a replacement figure that complies with the CC BY 4.0 license. Please check copyright information on all replacement figures and update the figure caption with source information. If applicable, please specify in the figure caption text when a figure is similar but not identical to the original image and is therefore for illustrative purposes only.

RESPONSE: It does not apply. We obtained permission from the original copyright holder to publish these figures.

Response to general Reviewers' comments:

Reviewer's Responses to Questions

Comments to the Author

1. Is the manuscript technically sound, and do the data support the conclusions?

Reviewer #1: Yes

Reviewer #2: Partly

Reviewer #3: Partly

RESPONSE: Thank you for the comment. We reviewed the manuscript and tried to improve it and focused the conclusions to the presented data. We changed it in the abstract and in the end of the discussion.

2. Has the statistical analysis been performed appropriately and rigorously?

Reviewer #1: Yes

Reviewer #2: Yes

Reviewer #3: Yes

RESPONSE: Thank you for the comment. 

3. Have the authors made all data underlying the findings in their manuscript fully available?

Reviewer #1: No

Reviewer #2: Yes

Reviewer #3: Yes

RESPONSE: We attached the databases in the submission.

4. Is the manuscript presented in an intelligible fashion and written in standard English?

Reviewer #1: Yes

Reviewer #2: Yes

Reviewer #3: Yes

RESPONSE: Thank you for the comment. 

5. Review Comments to the Author

Response to reviewer #1:

In this manuscript, the distribution of acquired syphilis, syphilis in pregnancy, and congenital syphilis was analyzed at the Brazilian border with other nine countries from 2010 to 2020. The results show that the number of cases for all three variables is increasing throughout the Brazilian border area. This is nothing new, as the number of syphilis cases has increased dramatically worldwide in recent years. However, given the lack of data on developing country borders, I believe that this manuscript should be considered for publication after some modifications as described below:

Abstract:

I suggest inserting a space between several words: pregnancy (SP), syphilis (CS), syphilis (AS), Brazil(38.4%), North(18.3%), South(65.7%), 427(72.6%), 441(75.3%), 422(72%).

RESPONSE: Thank you for the comment. We made the requested corrections in the text.

Introduction:

- Why "Discussions about the border have intensified in recent decades due to the regionalization processes"?

RESPONSE: We changed it. Now you can read the new explanation on page 2, lines 57 to 60 AND lines 63 to 65.

- About "Despite the importance of the border region in Brazil and the congenital syphilis as a public health problem, few studies explored the epidemiology of this infection in the region." Are AS and SP a public health problem in Brazil? Or not?

RESPONSE: Thank you for the comment. We added a paragraph in introduction. It is stated on page 3, lines 77 to 84.

Results:

- Note Figures 2 and 3. In the text below the figure, words are underlined. What does "2010 a 2020" mean?

RESPONSE: Thank you for the comment. We correct it, now it is 2010 to 2020.

- "Syphilis in pregnancy was reported in 61.4% (361) of border municipalities and 27.8% (164) have a detection rate ≥ national rate (21.6/1000 LB). Most municipalities, 67.5% (397), did not report cases of congenital syphilis in 2020 and 14.3% (84 municipalities ) have an incidence of congenital syphilis ≥ 7.7/1000 LB (Figure 5)." Define LB.

RESPONSE: Thank you for the comment. We wrote it down. LB = live births

Discussion:

- Italianize Treponema pallidum: "CS is the result of the transmission of the spirochete of the Treponema pallidum from the infected pregnant woman's ..."

RESPONSE: Thank you for the comment. We corrected it. 

Reviewer #2: 

The manuscript is of relevance, it points out updated aspects with the syphilis theme. However, it presents internal and external biases. 

Title: adequate; Abstract: incomplete with the need for adaptation of items as presented throughout the text; Introduction: presents issues that belong to another topic such as the research scenario, the knowledge gap is fragile, without articulation with the uniqueness of the research. 

RESPONSE: We rewrote the abstract to attend your suggestions. 

Methods: The research design is inadequate. According to Morgenstern (2011), ecological studies can be divided into: multiple group designs (exploratory and etiological), time trend designs (exploratory and etiological), mixed designs (between multiple groups and time trend, but also with sub-classifications between exploratory and etiological). By the detailing of the methods, the study is more articulated with an exploratory ecological study with multiple group designs. 

RESPONSE: Following your suggestion, we change the terminology of research design to be clear regarding what we did. 

Results: are very well designed, but not validated with explanatory tests. The statistics are simple and make the text uncompetitive. 

RESPONSE: We used secondary data and proposed to a descriptive exploratory study to have a general idea of the situation in the border areas. Even if we did not used exploratory tests, we think we have important data about syphilis in a region where there are not available data. We identified a problem to be approach by public health strategies in the border area. We agree that you had an important point and we intend to do it in a next proposal using primary data.

MORGENSTERN, Hal. Ecologic Study. In: ROTHMAN, Kenneth L.; GREENLAND, Sander; LASH, Timothy L. Modern Epidemiology, 3rd Edition, 2008.

Discussion: well elaborated with the inclusion of limitations. However, limited to the results, which in fact is not inconsistent, but does not address aspects related to the health model, the treatment of partners who are in another border country, the mobility of the standardization of treatment and availability of supplies among other aspects that interfere in the occurrence of the grievance in all its scopes, acquired syphilis, in pregnant women and congenital. I hope that the comments provided will help you with the publication elsewhere or to resubmit after adaptations.

RESPONSE: Thank you for the comment. We included this topic in the third paragraph of the discussion (page 13, lines 283-289). It is important to point out that we had difficulties for obtaining epidemiological information from neighboring countries.

Reviewer #3: 

Syphilis represents a highly relevant issue globally. It remains a global challenge, the second most commonly reported STI. In Brazil, syphilis persists as a public health problem, particularly due to limited access to timely diagnosis and treatment, as well as limited monitoring of cases in the Unified Health System health care network, especially in Primary Health Care. The challenge is amplified when the critical political and institutional moment of the country is recognized. One of the great challenges has been to implement these health care actions integrated with surveillance and control, ensuring wide access to diagnosis, treatment, and monitoring in the Primary Health Care setting. These aspects are even more critical in border areas in the context of South America.

There is great variation in the operational performance of disease control in the country. This variation has been associated with operational factors such as access to testing via rapid tests, but also to the lower use of condoms, the reduced use of penicillin in routine PHC, and the period in which there was a shortage of the drug. These aspects should have been better described in the manuscript.

RESPONSE: It is an important topic. We tried to be clear when explain it. You can read on page 16-17, lines 402-407.

Moreover, both acquired syphilis, syphilis in pregnant women, and congenital syphilis are compulsorily notifiable diseases in the country but have registered systematic under-reporting, which compromises health planning actions, despite the improvement over time. This aspect should have been studied in depth.

RESPONSE: It is an important point to be mentioned. We included it in the limitations of the study. We worked with the official data and included all the information available in the databases. It is stated on page 16, lines 389-391.

It is important to bring the impacts related to congenital syphilis to society.

For a broader look at the real epidemiological and operational situation of syphilis control, the analyses should include the quality of prenatal care in the public and private sectors. This aspect should be discussed.

RESPONSE: We agree with the reviewer that these are important topics, but it was not our goal to approach this question because this information is not included in the databases. It will be important to design a qualitative study or a quali-quantitative study to approach these situations. We plan to do it after the evaluation of the actual picture of the situation. We included it in the discussion section the impacts related to congenital syphilis to society and the importance of prenatal care evaluation. 

Expand the debate on the ethnic-racial clippings performed, as differentials in the three arches analyzed. Include the debates on recently published articles:

RESPONSE: Thank you for the comment. We included some of these references in our discussion.

Reiterate the relevance of PMAQ-AB as an innovative and useful action to induce quality improvement of PHC in SUS. This program was interrupted by the federal government in 2019 within the process of deconstruction of public health policies, having been replaced by the PREVINE Brazil Program. The authors need to discuss the effect of this change, implying a considerable setback in the process of evaluation and financing of PHC.

RESPONSE: Thank you for the comment. It is an important topic. As our data goes from 2010 to 2020 and the Previne Brasil was launched in the end of 2019 and it has been implemented in 2020 and then. We think the replacement of public health policies for primary care did not affect our data. Although we included a topic in the discussion section about the importance of the PMAQ-AB as a policy.

For the abstract, it is recommended to qualify the description of border arcs, to better situate the analyzed scenarios. 

RESPONSE: We tried to clarify the information about the arches, but we needed to explore more these data in the methods section because of the words limits. 

The objectives of the study should have been clearly listed, as in the introduction. Thus, the objectives of the study are partially articulated with a clear testable hypothesis stated. In principle, the focus is on the spatial and temporal description in Brazilian municipalities in border contexts. The conclusions of the abstract as well as of the manuscript should be adjusted to this perspective: the epidemiological and operational patterns of syphilis control are not satisfactory. The way it is described gives the wrong message of assumed control.

RESPONSE: Thank you for the comment. We adjusted the conclusions to the perspective of the objectives and tried to be clearer in the language.

The introduction needs to be enhanced with better-contextualized data in operational and epidemiological terms of syphilis control in the country and in border areas, especially over the period 2010 to 2020. In Brazil, the changes in the definition of syphilis case for compulsory notification purposes delimited the temporal scope of the study. It is important to signal which changes were undertaken and their impacts, particularly the change in the definition of appropriate treatment of pregnant women with syphilis, excluding as a criterion the concomitant treatment of the sexual partner, in terms of the sensitivity/specificity of the case definition criteria.

RESPONSE: We did not find published data about syphilis in pregnant women and congenital syphilis at the borders, and information about acquired syphilis was limited and very specific. Trying to improve the discussion, we included a phrase on page 13-14, lines 306-309

In the introduction, the authors should be clearer when referring to the fact that discussions about the border have intensified in recent decades due to regionalization processes, including more consistent references. The description of the study design needs to be qualified. In principle, the study design is appropriate to address the possible objective. In addition, a detailed map of the study area could have been presented, while in the text, the indication of the territorial and population, as well as the economic relevance of this territorial cut-out adopted in the study.

RESPONSE: We rewrote this topic to attend your suggestions. The new text is on page 2, lines 57-60 AND 63-65. The territories are described on the Methods Section and in Figure 1. 

It is recommended to detail the scope of the syphilis indicator panel with its linked databases, as well as the Health Information System for Primary Health Care, clearly demonstrating the role of care and surveillance and to what extent the interfaces between these systems. The population is clearly described and appropriate for the hypothesis being tested.

RESPONSE: The information contained in the indicators comes from the National Notification System, after performing the database linkage.

Qualify correctly and better the reference in the text to SPSS version 20.0 and TabWin (version???), according to the developers' specification. The correct statistical analysis is used to support conclusions.

RESPONSE: We included the version for the TABWIN. It was the 3.6. Added the SPSS and STATA were for windows.

There are concerns about ethical requirements being met.

RESPONSE: We did not understand why the reviewer has concerns about ethical requirements. The Project was submitted and approved by the Ethics Committee in the Federal University of Espirito Santo. We had added the approved letter to our submission. We can ensure you that we follow all the ethical procedures necessary to guarantee the quality and suitability of the data.

The results are clearly and completely presented. The figures (Tables, Images) are of sufficient quality for clarity.

RESPONSE: Thank you for the comment.

In the discussion, the reduction seen in the last two years analyzed does not allow one to clearly establish that there was a reduction in the detection rates analyzed. The limitations of the analysis are clearly described.

RESPONSE: Agreed. Thank you for the comment.

The authors discuss partially how these data can be helpful to advance our understanding of the topic under study. The public health relevance is addressed. However, the conclusions are partially supported by the data presented.

RESPONSE: We changed the conclusions and tried to be clearer regarding to the public health relevance. We tried to be clear when explain it. You can read on page 16-17, lines 402-407.

This article proposes to evaluate syphilis, a complex problem, through secondary data, in an extremely dynamic border scenario. To support improvements, changes in public policy, the first step is to know the situation through the available data. In this context we observe that syphilis in the region is a problem, the detection rates of the disease in pregnant women are higher than in other parts of Brazil, highlighting the importance of improving screening and access to diagnosis.

---

## [Decision Letter · Decision Letter 1]

13 Sep 2022

Gestational and congenital syphilis across the international border in Brazil

PONE-D-22-16035R1

Dear Dr. Miranda,

We’re pleased to inform you that your manuscript has been judged scientifically suitable for publication and will be formally accepted for publication once it meets all outstanding technical requirements.

Kind regards,

Everton Falcão de Oliveira, Ph.D

Academic Editor

PLOS ONE

Additional Editor Comments (optional):

Reviewers' comments:

Reviewer's Responses to Questions

**Comments to the Author**

1. If the authors have adequately addressed your comments raised in a previous round of review and you feel that this manuscript is now acceptable for publication, you may indicate that here to bypass the “Comments to the Author” section, enter your conflict of interest statement in the “Confidential to Editor” section, and submit your "Accept" recommendation.

Reviewer #1: All comments have been addressed

Reviewer #3: All comments have been addressed

2. Is the manuscript technically sound, and do the data support the conclusions?

Reviewer #1: Yes

Reviewer #3: Yes

3. Has the statistical analysis been performed appropriately and rigorously? 

Reviewer #1: Yes

Reviewer #3: Yes

4. Have the authors made all data underlying the findings in their manuscript fully available?

Reviewer #1: Yes

Reviewer #3: Yes

5. Is the manuscript presented in an intelligible fashion and written in standard English?

Reviewer #1: Yes

Reviewer #3: Yes

6. Review Comments to the Author

Reviewer #1: The authors have made all suggested changes, so the manuscript can be accepted for publication in Plos One.

Reviewer #3: The authors made the suggested changes, which improved this version. I have no additional suggestions for changes to the article.

7. PLOS authors have the option to publish the peer review history of their article (what does this mean?). If published, this will include your full peer review and any attached files.

Reviewer #1: **Yes: **Fred Luciano Neves Santos

Reviewer #3: No

---

## [Editor Report · Acceptance letter]

27 Sep 2022

PONE-D-22-16035R1 

Gestational and congenital syphilis across the international border in Brazil 

Dear Dr. Miranda:

I'm pleased to inform you that your manuscript has been deemed suitable for publication in PLOS ONE. Congratulations! Your manuscript is now with our production department. 

Kind regards, 

on behalf of

Dr. Everton Falcão de Oliveira 

Academic Editor

PLOS ONE